# Machine Learning Identifies Key Proteins in Primary Sclerosing Cholangitis Progression and Links High CCL24 to Cirrhosis

**DOI:** 10.3390/ijms25116042

**Published:** 2024-05-30

**Authors:** Tom Snir, Raanan Greenman, Revital Aricha, Matthew Frankel, John Lawler, Francesca Saffioti, Massimo Pinzani, Douglas Thorburn, Adi Mor, Ilan Vaknin

**Affiliations:** 1Chemomab Therapeutics Ltd., Tel Aviv 6158002, Israel; 2UCL Institute for Liver and Digestive Health, University College of London, London NW3 2PF, UK; 3Sheila Sherlock Liver Centre, Royal Free London NHS Foundation Trust, London NW3 2QG, UK; 4Department of Gastroenterology and Hepatology, Oxford University Hospitals NHS Foundation Trust, Oxford OX3 9DU, UK

**Keywords:** primary sclerosing cholangitis, proteomics, machine learning, clinical data, CCL24

## Abstract

Primary sclerosing cholangitis (PSC) is a rare, progressive disease, characterized by inflammation and fibrosis of the bile ducts, lacking reliable prognostic biomarkers for disease activity. Machine learning applied to broad proteomic profiling of sera allowed for the discovery of markers of disease presence, severity, and cirrhosis and the exploration of the involvement of CCL24, a chemokine with fibro-inflammatory activity. Sera from 30 healthy controls and 45 PSC patients were profiled with proximity extension assay, quantifying the expression of 2870 proteins, and used to train an elastic net model. Proteins that contributed most to the model were tested for correlation to enhanced liver fibrosis (ELF) score and used to perform pathway analysis. Statistical modeling for the presence of cirrhosis was performed with principal component analysis (PCA), and receiver operating characteristics (ROC) curves were used to assess the useability of potential biomarkers. The model successfully predicted the presence of PSC, where the top-ranked proteins were associated with cell adhesion, immune response, and inflammation, and each had an area under receiver operator characteristic (AUROC) curve greater than 0.9 for disease presence and greater than 0.8 for ELF score. Pathway analysis showed enrichment for functions associated with PSC, overlapping with pathways enriched in patients with high levels of CCL24. Patients with cirrhosis showed higher levels of CCL24. This data-driven approach to characterize PSC and its severity highlights potential serum protein biomarkers and the importance of CCL24 in the disease, implying its therapeutic potential in PSC.

## 1. Introduction

Primary Sclerosing Cholangitis (PSC) is a chronic, idiopathic liver disorder characterized by progressive fibrosis leading to cirrhosis, with more severe cases ending in hepatobiliary malignancies and liver failure requiring liver transplantation [1]. The pathology of PSC involves a ductular reaction and peribiliary inflammation with significant immune cell engagement, advancing to fibrosis, and in some cases, cirrhosis, significantly impairing liver function and increasing the risk of hepatobiliary cancers. Disease progression exacerbates patient morbidity and mortality, and despite the severe clinical outcomes associated with PSC, there are no approved drugs [2].

PSC can be challenging to assess, as the disease is heterogeneous and presents with variable progression, as is evident by the abundance of metrics and tests used by physicians to characterize the disease and monitor for disease progression, such as enhanced liver fibrosis (ELF) score, transient elastography (Fibroscan^®^), and magnetic resonance cholangiopancreatography (MRCP) [3]. More complex models also exist, aiming to combine several measurements into a single score, to predict outcomes. These include PRESTO, Mayo clinic PSC model, and the UK-PSC risk score. Novel biomarkers would potentially augment such models by incorporating relevant information and could improve their prognostic value. While invasive testing, such as liver biopsies, could offer valuable information, it is not a routine clinical practice given the patchy nature of PSC, limited value for disease staging, and the risk of damaging healthy tissue during the biopsy [4]. In contrast, serum-based biomarkers offer a safer and potentially more sensitive and specific alternative [5,6].

The data analyzed here were generated using a novel proximity extension assay (PEA) technology that allows for the quantification of multiple serum proteins with high sensitivity [7]. This is necessary for the discovery of potential biomarkers.

To analyze the proteomic data while accounting for its inherent complexities, such as high dimensionality and co-linearity of proteins, a suitable model is needed. In this work, we demonstrate the use of an elastic net regression model (ENRM), which can help prevent overfitting while maintaining model interpretability and has been used previously for similar tasks [8,9,10]. This model handles multicollinearity among predictors by combining the strengths of both ridge and Lasso (Least Absolute Shrinkage and Selection Operator) regression, applying a penalty to the model coefficients that balances the inclusion of significant variables and the exclusion of irrelevant ones. This approach facilitates the identification of a more precise set of proteins implicated in PSC pathogenesis and offers an understanding of disease mechanisms and potential therapeutic targets.

CCL24 (Eotaxin-2) belongs to a subgroup of CC chemokines that play a role in the immunopathogenesis of PSC [11,12]. CCL24 exhibits a high affinity for the CCR3 receptor on immune cells and fibroblasts, mediating their recruitment to sites of inflammation and their activation within the biliary epithelium, which leads to bile duct damage and fibrosis. The expression of CCL24 is induced in response to inflammatory stimuli in various cell types, such as endothelial cells, epithelial cells, and fibroblasts [13]. Several works have presented CCL24 as important for the understanding of PSC’s pathophysiology and as a potential therapeutic target aimed at modulating fibrosis and inflammation to delay or halt the progression of the disease [11,12]. CM-101, a first-in-class humanized antibody targeting CCL24, demonstrated efficacy in influencing consequential biomarkers of liver fibrosis in multiple preclinical models of PSC. We previously analyzed the serum proteomics of PSC patients and healthy controls (HC), revealing that key disease-related mechanisms are associated with CCL24 serum levels [12]. Revealed pathways, upstream regulators, and toxicity functions were elevated in PSC patients, particularly in patients with high CCL24 levels.

In this work, we demonstrate the role of computational analysis in identifying proteins of interest involved in PSC, offering a new perspective on proteins impacting PSC while emphasizing the need for novel biomarkers for PSC progression without invasive testing. The results presented here highlight specific proteins linked to PSC that could serve as biomarkers or potential therapeutic targets. Moreover, statistical modeling was employed to ascertain the presence of cirrhosis, utilizing proteins and pertinent clinical metrics.

## 2. Results

### 2.1. Proteomic Characteristics of the Studied Population

We compared the expression profile of serum proteins of two PSC cohorts with that of healthy controls (HC) (Figure 1A). Protein levels are expressed as normalized protein expression (NPX). When comparing serum levels of each protein, most proteins are over-expressed in patients with PSC compared to healthy controls (Figure 1B). Adjusted for multiple testing, 1628 proteins show a significant difference between conditions, with 135 proteins down-regulated and 1493 up-regulated in patients with PSC compared to HCs. When limiting the differentially expressed proteins to those with a fold change of 1.5 or more, three proteins are down-regulated and 72 proteins are up-regulated. The differential expression of all 2876 proteins can be found in Appendix A.

On average, patients with PSC were older (Table 1) than healthy controls, which likely resulted in some differences in the proteomic profile of the study population. However, a similar analysis on an age-matched subset of the data showed that age did not significantly affect the comparison [12]. Similarly, modeling expression values while adjusting for age as a covariate did not show an effect on the differential expression of proteins.

The proteomic profile can distinguish between the HCs and PSC patients, as seen in a PCA representation of the data (Figure 1C). As expected, due to the large number of proteins represented, some overlaps exist between the two groups. Taken together, this shows that protein levels in the serum of patients with PSC are associated with the disease.

### 2.2. Elastic Net Modeling

To uncover which proteins are most likely associated with the presence of PSC, an elastic net model was trained. This model was chosen based on its ability to account for the high dimensionality of our data and prevent overfitting. The trained model successfully predicted the disease state, with an accuracy of 0.98 (CI 0.96–1) and an AUC of 0.99 (CI 0.98–1). Top contributing proteins (Figure 2A) were estimated by repeating the modeling step 500 times and averaging the importance score, and proteins with an average importance of 1 or greater (*n* = 118) were considered relevant. A PCA plot (Figure 2B) was generated using this subset of the data and shows a strong separation of the population for disease presence.

The top proteins (Figure 2C) are all associated with cell adhesion, immune response, and inflammation, as can be expected. ROC curves were plotted for all proteins, with the top ones all having an area under the receiver operator characteristics (AUROC) curve of over 0.95 (Figure 2D), showing that these proteins are strong predictors of disease presence. The complete list of proteins that contributed to the model and their respective importance scores can be found in Appendix A.

### 2.3. Model Association with ELF Score

To focus the list of proteins on disease severity and increase the signature’s relevance to PSC, proteins were tested for their association with the ELF score, an indicator for fibrosis severity and predictor of PSC-related complications [14,15]. This score consists of a combination of three ECM-related markers: tissue inhibitor of metalloproteinases 1 (TIMP-1), amino-terminal propeptide of type III procollagen (PIIINP), and hyaluronic acid (HA). Out of all the proteins with considerable contribution (importance > 1) to the prediction, 16 also showed a strong (*p*-value < 0.05) correlation to the ELF score.

Using a heatmap (Figure 3A), strong clustering is seen for HC, PSC patients with low (<9.8) ELF scores, and PSC patients with high (>9.8) ELF scores. The same is seen for a principal component analysis (PCA, Figure 3B) showing how these three populations can be distinguished based on these proteins alone.

The proteins consisting of this signature belong to several categories, such as proteins that are bound to ECM components (ITGA5, ITGBL1, VWF, BCAM, and PKD1), cellular receptors (ADGRE2, ADGRE5, MSR1, DCBLD2, NOTCH3, HAVCR1, BCAM, and SRPX), and ECM structural components (LTPB2 and MFAP4). These functions are all relevant to PSC and its underlying mechanisms.

Using a threshold of 9.8 to stratify ELF score values (Table 2), ROC curves were generated for these proteins, with the top five all having an area under the ROC curve (AUROC) of over 0.8. This translates to the potential of these proteins to serve as biomarkers for both disease presence and severity, represented by the stratified ELF score.

For example, the protein with the highest importance that also had a strong correlation (0.82, Pearson, Figure 3D) with the ELF score, LTBP2, is overexpressed in the fibrotic livers of PSC patients [16] and was reported as a prognostic biomarker for liver cancer [17]. LTBP2 serves as a structural component of microfibrils without latent-TGF beta activity [18]. Interestingly, we previously demonstrated the therapeutic potential of blocking CCL24 in an Mdr2-knockout mouse PSC model [11] that was accompanied by reduced Ltbp2 expression within the biliary area following anti-CCL24 treatment (Appendix A).

### 2.4. Pathway Analysis

To translate the proteins driving the model into pathways, over-representation analysis (ORA) was performed. All proteins that contributed to the model in a meaningful way (importance over 1, *n* = 118) were used, resulting in 13 enriched pathways (Figure 4A). These pathways include those directly related to PSC and fibrosis, such as Elastic fiber formation (HSA-1566948), Integrin cell surface interactions (HSA-216083), and Extracellular matrix organization (HSA-1474244), as well as more general pathways for immune response and inflammation. We compared these pathways to those enriched in individuals stratified by CCL24 levels (*n* = 126, Figure 4B) and saw noticeable overlap between the two groups. (Figure 4C). The complete lists of proteins used for the over-representation analysis can be found in Appendix A for model contribution and in Appendix A for CCL24 stratification. The lists of enriched pathways can be found in Appendix A.

### 2.5. Statistical Modeling for the Presence of Cirrhosis

Next, we looked at the presence of cirrhosis as an indicator of progression to severe PSC (Table 3). Logistic regression adjusted for age showed several proteins and other clinical markers associated with the presence of cirrhosis. Patients with cirrhosis (*n* = 18) had lower levels of AST/ALT ratios and platelets (Figure 5A) and higher levels of CCL24, while other eotaxins (CCL11 and CCL26) did not show this pattern (Figure 5B). ROC curves show that these metrics and CCL24 are associated with the presence of cirrhosis (Figure 5C). The above-mentioned proteins that previously predicted disease state and showed a strong association with ELF, such as LTBP2, were not associated with the prediction of cirrhosis (Appendix A). Furthermore, when comparing patients who did not develop cirrhosis to those who were already cirrhotic at the time serum was taken and to patients who developed cirrhosis after sampling (Figure 5D), significant differences are shown for platelets (PLT) and CCL24. Finally, patients exhibiting elevated CCL24 levels were observed to have an increased probability of developing cirrhosis, as demonstrated by the Kaplan–Meier curve analysis (Figure 5D).

## 3. Discussion

This study shows the application of a machine learning approach integrated with proteomic data to highlight serum proteins associated with the presence and severity of PSC. We demonstrate that the circulating proteome differs among PSC patients, particularly those with a more severe disease, as indicated by their ELF score (>9.8) or the presence of cirrhosis. This observation, coupled with computational methods, allowed us to single out proteins of interest associated with the disease. Our iterative approach progressively refines the selection of a minimal protein signature for disease state, severity (using an ELF score threshold of 9.8), and cirrhosis.

The data used for this work, multiplex protein data obtained from a novel proximity extension assay, proved to be valuable in discerning disease severity; the breadth of these data, quantifying thousands of proteins, resulted in an accurate model and a thorough analysis of potential biomarkers. The dimensionality of the data was a major factor in model selection, as a relatively small sample size and feature abundance could result in overfitted or biased models. While more complex models exist, the exploratory nature of this research necessitated an interpretable model. The choice of an elastic net regression model, a regularization method that combines the properties of both ridge and lasso regression, resulted in a useful model that predicted disease presence while accounting for the large feature set, ensuring that the selected biomarkers are both statistically significant and clinically relevant.

The biomarkers identified through this study hold significant translational importance, offering potential advancements in the monitoring of disease progression in clinical settings. The utilization of proteomic data derived from blood tests, a minimally invasive method, underscores the beneficial potential of this approach in a clinical context where blood is taken regularly [19]. Moreover, the identified serum biomarkers could enhance the precision of disease monitoring, potentially leading to earlier interventions, tailored treatment plans, and improved patient outcomes. The integration of these biomarkers into clinical studies could therefore benefit how diseases are tracked over time.

A machine learning-based model identified a protein signature that can serve as biomarkers of PSC and its progression, and statistical modeling identified biomarkers that are elevated in PSC-related cirrhosis. However, proteins that correlated with the development of cirrhosis, such as CCL24, are not biomarkers per se, but their elevated levels indicate a role in disease pathogenesis. This is consistent with a previous report of increased CCL24 levels in hepatitis B patients who developed cirrhosis [20].

Several studies have indicated the involvement of CCL24 in PSC [11,12]. CCL24 is up-regulated in PSC patients and could be involved in the recruitment of immune cells to the liver and bile ducts and activation of hepatic stellate cells and cholangiocytes, contributing to the inflammatory-fibrotic-cholestatic process characteristic of PSC. The pathway analysis described in this work underscores the intricate involvement of various biological pathways in PSC, ranging from fibrotic-related pathways to pathways associated with immune response and inflammation. Notably, the comparison of enriched pathways between individuals stratified for CCL24 revealed a significant overlap, further strengthening the potential role of CCL24 in modulating the disease.

This work summarizes our research on a population of patients with PSC compared to those without the disease. PSC is a rare disease which allowed only a limited patient population to be studied, and repeating this study would be necessary. This also comes into effect when modeling the data, as a small sample size could lead to overfitting. While our choice of algorithm considered the limitations of the data, more robust approaches are constantly being developed [21,22] and could lead to strong results once employed. Finally, the exploratory nature of this work would benefit from validation in a clinical setting to observe how specific protein levels or clinical measurements change over time.

## 4. Methods

### 4.1. Study Population

This study included three distinct groups: two patient cohorts diagnosed with PSC and a control group consisting of healthy individuals. The first PSC cohort comprised 30 participants from whom serum samples were obtained at the UCL Institute for Liver and Digestive Health, Royal Free Hospital, London, UK. This cohort had Enhanced Liver Fibrosis (ELF)™ score for 20 out of 30 participants and records of whether a patient developed cirrhosis or not. The second PSC cohort consisted of 16 individuals providing baseline serum samples, all of whom were participants in the SPRING phase 2a clinical trial (NCT04595825), exploring the safety and biological effects of CM-101 in PSC patients. Of these, 13 out of the 16 had ELF scores available. The control group involved 30 healthy participants, with serum samples collected pre-dose from a phase 1 clinical trial (NCT06025851), assessing the safety and tolerability of CM-101. No ELF scores were available for these healthy controls.

The collection of samples was conducted in strict accordance with the ethical guidelines of the Declaration of Helsinki, and the requisite approvals were obtained from the ethics committees of the involved institutions. Written consent was secured from all subjects before collecting the samples. Demographic and clinical characteristics, including age, gender, serum biochemistry, complete blood counts, and any indicators of cancer or symptoms related to the disease, were collected and recorded for analysis. The complete demographic and baseline characteristics by cohort is available in Table 1, and alternative stratification for ELF scores is available in Appendix A. For patients in cohort 1, a demographic table stratified according to the presence of cirrhosis appears in Table 3.

### 4.2. Serum Proteomics Assay

Using the Olink^®^ Explore 3072 platform, we assessed serum levels of 2926 individual proteins through the proximity extension immunoassay (PEA; Olink^®^ Proteomics, Uppsala, Sweden). This method employs a pair of antibodies tagged with complementary DNA to detect proteins in serum. The reaction was amplified and quantified using PCR, producing normalized protein expression values (NPX) on a log-2 scale that allowed for relative quantification across samples. Following internal quality control from Olink, 2876 proteins were used in the downstream analysis. Serum levels of CCL24 were found to be strongly correlated with its NPX values. (Appendix A). 

The analysis was conducted using the R statistical software (version 4.3.0; R Core Team 2023 [23]), during which data underwent preprocessing, including the recoding of identifiers. An outlier, identified due to abnormally high protein levels possibly from technical artifacts, was removed from further analysis. This decision was supported by principal component analysis (PCA) and QC plots generated with the Olink^®^ Analysis package in R (version 3.4.1) and the functions “olink_pca_plot” and “olink_qc_plot”, respectively. Other than this exclusion, all data were retained, including NPX values beneath the detection threshold and those flagged by QC alerts.

The study population was stratified into patients with PSC and HC. Further, patients were subgouped by fibrosis severity, diagnosed by ≥9.8 (high fibrosis) or ELF score < 9.8 (Low fibrosis) [24]. Differential expression between each group was calculated using a Welch 2-sample *t*-test, corrected for multiple testing using the Benjamini–Hochberg (BH) method, with a *p*-value of 0.05 or lower considered statistically significant. Data were processed using the tidyverse (version 2.0.0, [25]) package suite. An annotated volcano plot of the 30 most differentially expressed proteins between PSC and HC can be found in Appendix A.

### 4.3. Elastic Net Modeling

All 2870 proteins were used as features, with the binary class of PSC or HC used for the prediction. Modeling was implemented in R using the glmnet R package (version 4.1–7, [26]) via the caret (6.0–94, [27]) package. To minimize overfitting, repeated 5-fold cross validation was performed 10 times using an 80/20 split for training and testing. The model was tuned for alpha and lambda ranging from 0.1 to 0.9 in 0.1 increments, and 0 to 1 in 0.01 increments, respectively. To evaluate the diagnostic performance, receiver operating characteristic (ROC) curves were constructed, and areas under ROC curve (AUC) were calculated using the R package pROC (1.18.5, [28]).

### 4.4. Pathway Analysis

Pathway analysis and over-representation analysis were performed using the Search Tool for Retrieval of Interacting Genes/proteins (STRING, [29]) database, with default settings. The proteins used for enrichment in PSC patients were those that contributed to the prediction, with an importance of 1 or greater (*n* = 118). For CCL24, patients were stratified by the median value of CCL24, and a *t*-test was performed, with proteins with a *p*-value of 0.05 or lower included in the over-representation analysis (*n* = 126). Pathways were considered significantly enriched with a false discovery rate (FDR) of 0.05 or lower, corrected for multiple testing using the BH method. Strength is the log10 (observed/expected). The complete result of the STRING query can be found in Appendix A.

### 4.5. Statistical Modeling

To assess the presence of cirrhosis in patients, a logistic regression was fitted using the following equation:$$\log\left(\frac{p}{1-p}\right)=\beta_{0}+\beta_{1}X_{1}+\beta_{2}X_{2}+\ldots +\beta_{n}X_{n}+\beta_{\mathrm{age}}\mathrm{Age}$$
where *p* is the probability of cirrhosis, $\beta_{0}$ is the intercept, $\beta_{1}$, $\beta_{2}$,…, $\beta_{n}$ are the protein levels and numeric clinical metrics, and $\beta_{age}$ is the coefficient for the age variable. The model was adjusted due to the difference in age between groups (Table 3).

### 4.6. Animals

Treatment of Mdr2-knockout mice with CM-101 and transcriptomic analysis of the biliary area was described previously [11]. Briefly, 6-week-old mice were subcutaneously treated with 10 mg/kg CM-101 for 6 weeks. Paraffin-embedded sections were analyzed using the whole mouse transcriptome atlas (NanoString, Seattle, WA, USA).

## 5. Conclusions

This research employed an iterative approach to identify circulating biomarkers for PSC and assess their correlation with disease severity, yielding a minimal protein signature with diagnostic utility for PSC, fibrosis progression, and identification of cirrhosis. Notably, CCL24 displayed elevated expression in cirrhotic patients, strengthening its involvement in PSC progression.

## Figures and Tables

**Figure 1 ijms-25-06042-f001:**
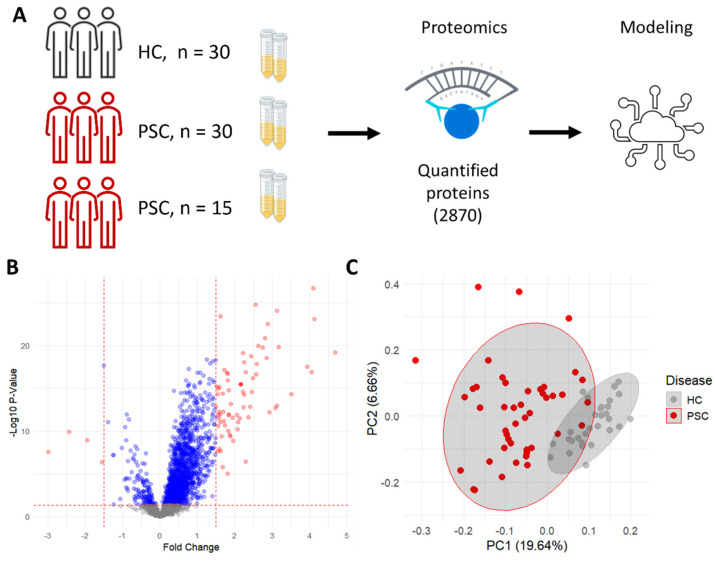
Proteomic profiling demonstrates unique protein expression in PSC patients. (**A**) In the data acquisition process, serum was taken from three cohorts of individuals, quantified by the real-time PCR-based proximity extension assay (PEA) and used to model disease presence. (**B**) Volcano plot showing the fold change (red points represent FC > 1.5, blue points represent FC < 1.5) and −Log10 of the *p*-value from a Welch two-sample *t*-test between HC and PSC patients. (**C**) Principal component analysis showing the separation of HC and PSC patients.

**Figure 2 ijms-25-06042-f002:**
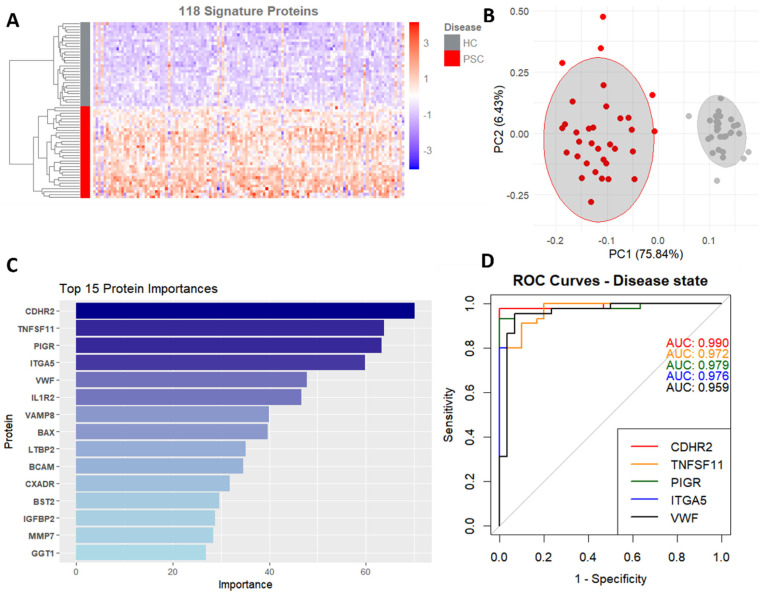
Elastic net logistic regression identifies disease biomarkers with high predictive values. (**A**) Heatmap based on 118 proteins contributing to disease presence prediction. (**B**) PCA using the same 118 proteins. (**C**) Protein importance for the top 15 proteins, averaged across 500 runs. (**D**) ROC curves and area under the curve for the top five proteins predicting disease state.

**Figure 3 ijms-25-06042-f003:**
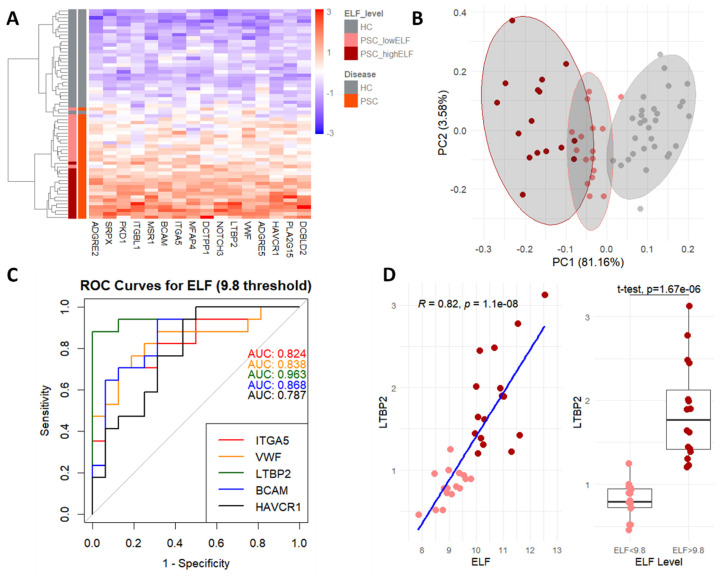
Association of model to disease severity. (**A**) Heatmap of protein signature for PSC and ELF score, showing hierarchical clustering of HC and patients with PSC with low (<9.8) and high (>9.8) ELF scores. (**B**) PCA using only the proteins used for the heatmap. (**C**) ROC curves and area under the curve for the top five proteins predicting both disease and ELF score. (**D**) Correlation and *t*-test of LTBP2 and ELF, both as a continuous variable and stratified by 9.8 threshold.

**Figure 4 ijms-25-06042-f004:**
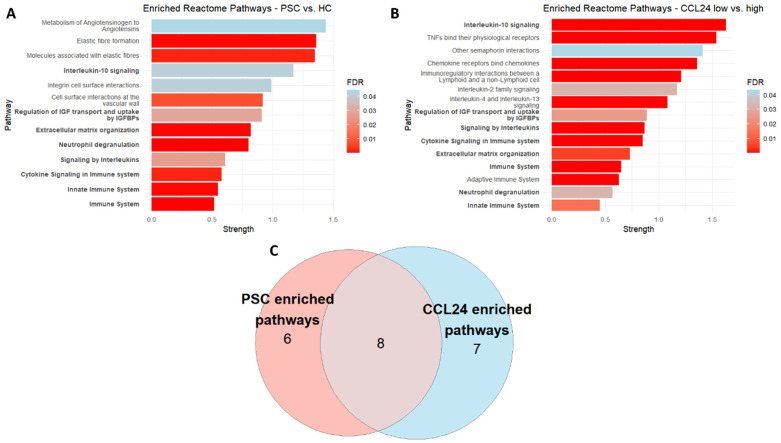
CCL24 is associated with disease-related biological pathways. (**A**) Reactome pathways enriched for PSC/HC. (**B**) Reactome pathways enriched for CCL24 high/low (**C**) Venn diagram of the overlap between pathways enriched for proteins that contributed to the PSC/HC model and proteins that were differentially expressed in PSC patients with high/low levels (by median NPX value) of CCL24.

**Figure 5 ijms-25-06042-f005:**
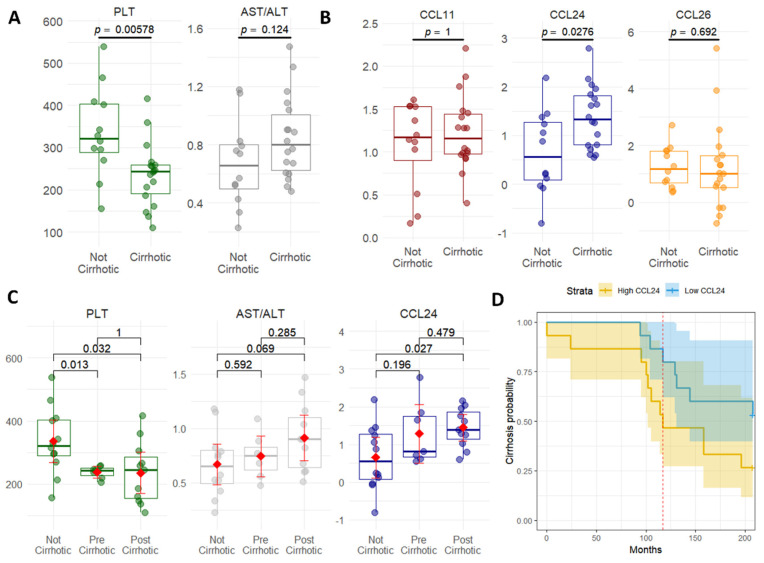
Changes in CCL24 expression reflect cirrhosis presence. (**A**) Wilcoxon test comparing the mean values of platelets and AST/ALT ratio between PSC patients with and without cirrhosis. (**B**) Wilcoxon test comparing the mean values of CCL11, CCL24, and CCL26 between PSC patients with and without cirrhosis. (**C**) Wilcoxon test comparing the mean values of PLT, AST/ALT, and CCL24 between PSC patients without cirrhosis, those who were cirrhotic when serum was taken, and those who developed cirrhosis after serum was taken. Red dot and line represent the mean value. (**D**) Kaplan–Meier plot stratified by median CCL24 levels showing probability of cirrhosis over time.

**Table 1 ijms-25-06042-t001:** Demographic information per cohort.

		Cohort		
Characteristic	N	HC, N = 30 ^1^	PSC, Cohort 1, N = 30 ^1^	PSC, Cohort 2, N = 15 ^1^
ELF_level	75			
HC		30 (100%)	0 (0%)	0 (0%)
PSC, ELF < 9.8		0 (0%)	8 (27%)	8 (53%)
PSC, ELF > 9.8		0 (0%)	12 (40%)	4 (27%)
PSC, ELF NA		0 (0%)	10 (33%)	3 (20%)
Age	75	24 (21, 28)	46 (32, 65)	37 (30, 53)
Gender	75			
Female		0 (0%)	11 (37%)	8 (53%)
Male		30 (100%)	19 (63%)	7 (47%)
ELF	32	NA (NA, NA)	10.03 (9.30, 10.73)	9.41 (8.96, 10.09)
Unknown		30	10	3
ALP	75	74 (58, 87)	235 (150, 398)	282 (259, 460)
AST	75	19 (18, 24)	48 (33, 75)	73 (39, 95)
ALT	75	17 (13, 23)	64 (45, 153)	98 (42, 207)
Fibroscan	43	NA (NA, NA)	10.20 (8.65, 11.60)	8.10 (6.80, 10.70)
Unknown		30	0	2

^1^*n* (%); Median (IQR).

**Table 2 ijms-25-06042-t002:** Demographic information per ELF score category.

Disease and ELF State						
Characteristic	N	Overall, N = 75 ^1^	HC, N = 30 ^1^	PSC, ELF < 9.8, N = 16 ^1^	PSC, ELF > 9.8, N = 16 ^1^	PSC, ELF NA, N = 13 ^1^
Age	75	31 (25, 51)	24 (21, 28)	31 (25, 35)	54 (47, 64)	49 (35, 68)
Gender	75					
Female		19 (25%)	0 (0%)	7 (44%)	11 (69%)	1 (7.7%)
Male		56 (75%)	30 (100%)	9 (56%)	5 (31%)	12 (92%)
ELF	32	9.88 (9.02, 10.39)	NA (NA, NA)	9.01 (8.79, 9.37)	10.49 (10.10, 11.08)	NA (NA, NA)
Unknown		43	30	0	0	13
ALP	75	150 (76, 287)	74 (58, 87)	287 (251, 361)	296 (196, 584)	203 (76, 282)
AST	75	33 (20, 67)	19 (18, 24)	67 (36, 88)	52 (42, 79)	42 (31, 72)
ALT	75	38 (19, 96)	17 (13, 23)	105 (57, 188)	64 (50, 160)	46 (33, 154)
Fibroscan	43	10.10 (7.95, 11.60)	NA (NA, NA)	8.45 (7.33, 11.60)	10.30 (9.00, 12.20)	9.85 (8.48, 10.40)
Unknown		32	30	0	1	1

^1^ Median (IQR); *n* (%).

**Table 3 ijms-25-06042-t003:** Demographic information of cohort 1 according to cirrhosis presence.

			Cirrhosis		
Characteristic	N	Overall, N = 30 ^1^	N, N = 12 ^1^	Y, N = 18 ^1^	*p*-Value ^2^
Age	30	46 (32, 65)	32 (30, 50)	53 (37, 67)	0.072
Gender	30				>0.9
Female		11 (37%)	4 (33%)	7 (39%)	
Male		19 (63%)	8 (67%)	11 (61%)	
ELF	20	10.03 (9.30, 10.73)	9.01 (8.72, 10.74)	10.09 (9.91, 10.45)	0.2
Unknown		10	4	6	
ALP	30	235 (150, 398)	244 (155, 330)	235 (150, 535)	>0.9
AST	30	48 (33, 75)	70 (34, 120)	47 (34, 56)	0.2
ALT	30	64 (45, 153)	162 (76, 198)	56 (42, 69)	0.014
Fibroscan	30	10.20 (8.65, 11.60)	10.10 (9.20, 11.68)	10.30 (8.65, 11.38)	0.9
CCL11	30	1.17 (0.97, 1.48)	1.17 (0.90, 1.53)	1.16 (0.97, 1.44)	>0.9
CCL24	30	1.15 (0.60, 1.63)	0.56 (0.08, 1.27)	1.33 (0.81, 1.82)	0.028
CCL26	30	1.06 (0.53, 1.76)	1.18 (0.68, 1.81)	1.02 (0.52, 1.63)	0.7

^1^ Median (IQR); *n* (%); ^2^ Wilcoxon rank sum test; Fisher’s exact test; Wilcoxon rank sum exact test.

## Data Availability

The serum proteomics data are not publicly available, as these data are part of an ongoing clinical trial. The data are available from the corresponding author upon reasonable request.

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
