# Peer review of "Machine Learning Identifies Key Proteins in Primary Sclerosing Cholangitis Progression and Links High CCL24 to Cirrhosis"

_ijms, 2024, doi:10.3390/ijms25116042_

Round 1

Reviewer 1 Report

Comments and Suggestions for Authors

This Article: Machine Learning Identifies Key Proteins in Primary Sclerosing Cholangitis Progression and Links High CCL24 to Cirrhosis, The authors identified CCL24 as a serum biomarker of PSC. 

Data modeling was performed using conventional R packages applied to proteomics data.  Significant up- and down-regulated predictors were identified in longitudinal studies, derived from ongoing clinical trials. 

-It would be helpful to readers to understand what molecules/proteins cause the upregulation of CCL24 (e.g. upstream in the pathways), and if those upstream regulators are also present in the peripheral blood.  

- Is there any difference in medications between PCS and non-PSC groups?

-Could you please justify why using an enhanced liver fibrosis (ELF) score is appropriate, to define a protein signature?

- Fig 4 A, B: please comment on why the IL-10 pathway is enriched in the CCL24 Low group. How many molecules (out of how many known in the pathway) are up- or down-regulated? 

Author Response

We thank the reviewer for his comments, and responded to each as follows:

Regarding the upstream regulators of CCL24, this was explored in our recent publication (Greenman R et al. Cells. 2024. PMID: 38334601, Supp. Figure S1C) and as such was not explored in the current narrative, and only mentioned briefly in the introduction: “We previously analyzed serum proteomics of PSC patients and healthy controls (HC), revealing that key disease-related mechanisms are associated with CCL24 serum levels. Revealed pathways, upstream regulators and toxicity functions were elevated in PSC patients, particularly in patients with high CCL24 levels”. As the reviewer suggests, these proteins are present in the blood, and were shown to have low levels in controls compared to PSC patients, and mostly higher in PSC patients with high CCL24 levels.

Regarding medication taken by the study population, healthy controls and PSC cohort 2 were not taking any medication. PSC cohort 1 was taking standard care medications. While we acknowledge that a different treatment regimen could influence serum protein levels, the origin of the data prevents us from controlling for this factor. A comparison between the two PSC cohorts did not demonstrate a distinct difference in proteomic profile, which led us to conclude that this potential bias did not substantially alter the results. 

Regarding the use of ELF score for stratification, ELF score is a widely accepted estimator for liver fibrosis severity. Whereas in other chronic liver diseases fibrosis is more commonly staged by biopsy, in a patchy disease such as PSC, the serological ELF score can perform as good or better than histological scoring (Muir AJ et al. Hepatology. 2019. PMID: 30153359.). Therefore, the ELF score is used to identify a fibrosis signature (Ramnath D et al. JCI Insight. 2018. PMID: 30046009; Irvine KM et al. PLoS One. PMID: 27861569). We revised the text to clarify this point, both in the Methods section (“Further, patients were subgouped by fibrosis severity, diagnosed by >=9.8 (high fibrosis) or ELF score <9.8 (Low fibrosis)”), and in the Results section(“To focus the list of proteins on disease severity and increase the signature’s relevance to PSC, proteins were tested for their association with the ELF score, an indicator for fibrosis severity and predictor of PSC-related complications”.

Regarding the relevance of IL-10 pathway in CCL24 stratified groups, while the connection is relatively understudied, it seems that IL-10 is a positive regulator of CCL24, when combined with IL-4 (Makita N et al. Int Immunol. 2015. PMID: 25267883). We previously identified IL-10 as an upstream regulator of CCL24 in PSC patients and reported that result in a recent publication (Greenman R et al. Cells. 2024. PMID: 38334601). The results of the over-representation analysis for the CCL24 stratified groups can be found in supp. table 4, which also includes the number of proteins that were found to be differentially regulated out of the full pathway.

Reviewer 2 Report

Comments and Suggestions for Authors

I was highly interested to read this study from Snir and colleagues on the application of machine learning for identifying protein biomarkers of PSC progresion.

The manuscript is well-written, the methods are explained in details and the results are cearly presented and the conclusions are supported by the results.

I only have some minor comments. First, that the control group has not involved any women, so this may affect statistical analysis. In this context I believe that the age-matche sub-analysis made for the proteomic characteristics refers to male patients.

Second, it is not very clear to me the stratification of patients according to CCL24 levels. In this stratification only PSC patients are included or also the HC? Moreover, I understand that CCL24 has been shown to paly an important role in PSC pathogenesis in previous studies, however in not among the top differentailly expressed proteins in this study unlike LTBP2, which also has the best correlation with ELF score. So, I do not know if LTBP2 instead of CCL24 in the statistical modeling for the presence of cirrhosis could perform equaly or better.

Overall is a well-designed, interesting and very-well presented work.

Author Response

We thank the reviewer for his comments, and responded to each as follows:

Regarding the effect of gender on the analysis, the reviewer is correct in suggesting that gender introduces some bias to the analysis since the healthy control group consists of males only. Broadly, each protein is affected to an extent by gender, as can be seen by Olink’s own analysis on a large population (https://insight.olink.com/data-stories/normal-ranges, see representative figure below for CCL24 and several other proteins). Correcting this effect for each protein based on our limited sample is not feasible, and when looking at a PCA plot of the entire protein profile of patients with PSC, no noticeable differences appear between genders. This, combined with the fact that our control group was taken from a phase 1 clinical trial which is usually done on males only, we believe that the potential bias is minimal. 

Regarding the comment on CCL24 stratification, we clarified that it was done on the PSC patients’ population only. This is to avoid the inherent bias in protein levels of this population compared to healthy controls. The language was made clearer to reflect this.

Regarding our use of CCL24 over LTBP2 in the presence of cirrhosis, statistical modelling was performed on all proteins, including LTBP2 and the rest of the protein signature described in the work. We added the result of this model to supplementary table 5. These other proteins were not associated with cirrhosis. Cirrhosis is an end-stage complication of PSC. The fact that CCL24 is higher in patients that have already developed cirrhosis suggests that CCL24 is not a biomarker per se, but its elevated levels indicate a role in disease pathogenesis.

We rephrased the results to clarify that the proteins of the signature set did not predict cirrhosis: “The above-mentioned proteins that previously predicted disease state and showed a strong association with ELF, such as LTBP2, were not associated with prediction of cirrhosis.”